# Food and nutrient intake among non-pregnant, non-lactating women of reproductive age of Mbeya in Tanzania: A repeated 24-hour dietary recall

Nyamizi J. Ngassa[1], Ray M. Masumo [1]*, Adam Hancy[1], Esther J. Kabula[1], Erick Killel[1], Jackline Nusurupia[1], Adeline Munuo[1], Hoyce Mshida [1], Rose Mpembeni [2], Elizabeth Lyimo[1], Rose V. Msaki[1], Theresia Jumbe[3], Germana H. Leyna[1,2]

1 Tanzania Food and Nutrition Centre (TFNC), Dar es Salaam, Tanzania, 2 Muhimbili University of Health and Allied Sciences (MUHAS), Dar es Salaam, Tanzania, 3 Sokoine University of Agriculture (SUA), Morogoro, Tanzania

* rmasumo@yahoo.com

**Data Availability Statement:** Available data here belong to the authors, URL: https://www.

## Abstract

Micronutrient deficiencies can hurt the health of women of reproductive age (WRA), their pregnancy outcomes, and the growth and development of their offspring in sub-Saharan African countries. The present study aimed to estimate the dietary intake of non-pregnant and non-lactating (NPNL) WRA, residing in seven districts of the Mbeya region in Tanzania using a 24-hour dietary recall. A cross sectional study was conducted among 500 NPNL WRA. A two-stage sampling method was used, three enumeration areas (EAs) were chosen for each district using the probability proportional to size and, 24 eligible households were randomly selected from each chosen EA. The SAS macros (version 9.4) were used to analyse the quantities consumed and explore the usual intakes of three fortified food vehicles. The median intake of oil, wheat flour, and calories was 36.47g, 110.53g, and 2,169.9 kcal, respectively. The median protein intake was 63.5g, higher than the recommended value of 56.0g. Among the micronutrients, zinc had the highest prevalence of inadequate intake (91.2%), followed by iron (82.2%), and vitamin B12 (80.0%). Vitamin E had the highest nutrient gap (50.7%), while the nutrient gap in Q1 was relatively low (9.8%). There was a moderate prevalence of inadequate intake of vitamin C (46.5%) and riboflavin (54.8%), with a nutrient gap in Q1 (42%). Inadequate intake of vitamin A, thiamine, niacin, vitamin B6, and folate was moderate, ranging from 32.6% to 44.4%, with a nutrient gap at Q1 ranging from 16.2% to 34%. Our study has revealed a prevalent inadequate intake of micronutrients among NPNL WRA. Furthermore, our findings suggest that fortifying oil represents a promising intervention. However, fortified wheat flour had limited reach among NPNL WRA in the Mbeya region of Tanzania.

advancingnutrition.org/sites/default/files/2023-09/usaid_an_intake_survey_tz_2023.pdf.

**Funding:** This work was supported by the United States Agency for International Development (USAID) through USAID Advancing Nutrition. The funding was provided under contract number 7200AA18C00070, awarded to JSI Research & Training Institute, Inc., with a subgrant allocated to the Tanzania Food and Nutrition Centre (TFNC) for GHL. The funders had no role in study design, data collection and analysis, decision to publish, or preparation of the manuscript.

**Competing interests:** The authors have declared that no competing interests exist.

## Introduction

Micronutrient deficiencies can have adverse effects on the health of women of reproductive age (WRA), their pregnancy outcomes, growth, and the development of their offspring [1]. WRA are particularly susceptible to micronutrient deficiencies due to inadequate dietary intake, unequal distribution of food within the household, societal norms, and gender-based discrimination that expects women to prioritize their family's nutritional needs over their own [2].

It is estimated that around three billion people worldwide cannot afford to maintain a healthy diet, as reported in 2021 [3]. The consumption of low-micronutrient diets is a significant contributor to micronutrient deficiencies, which persist as a major challenge in developing countries such as Tanzania [4]. Micronutrient deficiencies, particularly with regards to iron, vitamin A, iodine, and zinc, are among the top ten leading causes of morbidity and mortality among WRA in low-income countries [5, 6]. These deficiencies can have long-term consequences on health, learning ability, and productivity. Iron Deficiency Anaemia (IDA), Iodine Deficiency Disorders (IDD), and Vitamin A Deficiency (VAD) are the most common micronutrient deficiencies reported in Tanzania, according to the Tanzania Demographic and Health Survey (TDHS) [7]. Other micronutrient deficiencies of concern in Tanzania include Zinc, Folic Acid, Calcium, and some B vitamins (B12 and niacin), as reported by the TDHS [8]. In Tanzania, the prevalence of anaemia among WRA (15–49 years) was reported to be 45% with pregnant women having a much higher prevalence (57%) [9, 10]. Furthermore, the TDHS reported that 36% of WRA in Tanzania had Vitamin A deficiency. Although the role of Vitamin A deficiency in causing anaemia is not fully established, it may affect red blood cell production [11].

WRA make up 48% of the total female population in Tanzania, and the total fertility of 4.39 [7, 12]. Energy and protein shortages, along with micronutrient deficiencies, are common among non-pregnant and non-lactating women (NPNL), posing serious public health concerns [1]. In Africa, literatures have shown that dietary behaviors are influenced by cultural, geographical, and food preferences [13–18]. Additionally, women from socially and economically disadvantaged backgrounds are more affected by these issues [14–19]. Due to the limited coverage and longer periodicity of national nutritional surveillance in the country, ad hoc surveys offer a quick way to evaluate the diets of targeted populations.

There is a paucity of information about the food and nutrient intake in most sub-Saharan African countries, including Tanzania [6, 13, 15, 19]. This information is important to understand energy intake, as well as the intake of macronutrients (fats, proteins, and carbohydrates), micronutrients (selected vitamins and minerals), food groups, and food fortification. With this knowledge, policies and programs can be informed with new evidence to improve nutrient intake, dietary patterns, and the overall nutritional status of the population [6, 13, 15, 19]. Therefore, the present study aimed to estimate and describe the distribution of food and nutrient intake among WRA 15–49 years who are NPNL in seven districts of the Mbeya region in Tanzania.

## Materials and methods

### Ethical statement

This study obtained ethical permit from the National Health Research Ethics Review Committee (NatHREC) at National Institute for Medical Research with reference no. NIMR/HQ/R.8a/Vol.IX/3964. All study participants provided written consent prior to participating in the study and, participants did not receive financial compensation for participating. For the

participants under 18 years of age, written informed consent was also obtained from their parents/guardians. All data were anonymised and, will be destroyed when the project is concluded.

## Study design and study population

A cross-sectional study was conducted in the Mbeya region of Tanzania from July to September 2022. The study enrolled NPNL WRA aged 15 to 49 years old from all seven districts of the region, namely Chunya DC, Mbeya DC, Mbeya City, Mbarali DC, Kyela DC, Rungwe DC, and Busokelo DC.

## Sample size and sampling procedures

To conduct a statistically valid study, a pre-calculated representative sample size of 504 NPNL WRA (15 to 49 years) approached, 500 agreed to participate (total response rate 99.2%). The sample size of 500 was enough based on the pre-calculated sample size using the Lwanga and Lemeshow formula [20]. This calculation assumed a prevalence of 50%, a 5% margin of error, a 95% confidence level, and a design effect of 2. An additional 10% was added to account for potential non-participation or communication difficulties due to illness. Residents in refugee camps were excluded from the study since they did not qualify as permanent residents of the Mbeya region.

To select participants for the study, a two-stage sampling method was used. In the first stage, three enumeration areas (EAs) were chosen for each district using the probability proportional to size (PPS) method. In the second stage, 24 eligible households were randomly selected from each chosen EA. This was based on a completed list of all households within the EA. Before data collection began, the study team worked with local leaders to list all eligible households in the area. In each designated household, a single NPNL WRA between 15 and 49 years was purposefully selected to participate in the survey. A subset of 100 participants, comprising 20% of the study population, was identified for a repeated 24-hour dietary recall. Furthermore, approximately five participants per EA were randomly enlisted for a repeated recall within 3 to 8 weekdays after the initial recall [21–23].

## Data collection

Fourteen researchers trained in data collection gathered the information using a household questionnaire for NPNL WRA, a 24-hour dietary recall questionnaire, and an anthropometry form. The household module obtained demographic information for each respondent and their household. We administered a 24-hour dietary recall questionnaire to assess information on foods and drinks consumed. A tablet-based questionnaire programmed into the Open Data Kit via the Kobo toolbox and imported it into Excel and Stata for cleaning and analysis. Paper forms were also available. The standard questionnaire was translated from English to Swahili and back-translated for accuracy by different translators. Swahili serves as the official language of Tanzania and is spoken with proficiency by over 95% of Tanzanians. The completed Swahili questionnaire underwent a meticulous review process to ensure the accuracy and cultural appropriateness of its content, aligning it conceptually with the original version. Subsequently, the questionnaire was administered to 57 NPNL WRA 15 to 49 in Ubungo District, Dar es Salaam, yielding a response rate of 95%. This meticulous evaluation aimed to gauge the quality of translation in terms of comprehensibility, readability, and its relevance in assessing face validity.

## Data cleaning and preparation

The researchers converted the portion sizes of food consumed into grams by multiplying the portion consumed by the measured weights of the foods listed in the photobook created for the study [21]. If the food wasn't listed in the photobook, they multiplied the reported volume of consumption by density values published by the Food and Agriculture Organization (FAO) and the United States Department of Agriculture (USDA).

## Food matching, data quality control and validation

For the 66 distinct single foods extracted from recipes, we calculated the ingredient fraction for each recipe ingredient to estimate the consumed quantity of the cooked single food, and reported total amount of the recipe consumed. We matched the 71 single foods directly reported (e.g., avocado, rice eaten raw, or sugar) with items included in the FCT on a 1:1 basis, and the quantities reported in the survey were taken as the quantities consumed. We developed a Visual Basic for Applications (VBA) Excel macro to automate the finding of each reported single food name and compute its corresponding consumed quantity. These single foods were then matched with four food composition tables, two for Tanzania (one with non-fortified oil and wheat flour and one with fortified values), and two for Uganda (one without and one with fortified values). We matched new foods not documented in the TFCT with food composition data of Kenya, Malawi, or the United States as appropriate. A comprehensive, up-to-date TFCT for the Tanzanian diet was not available when this study commenced. The use of multiple food composition tables was designed to enrich the TFCT. The multiple FCT served as a practical resource to ensure that the study covered a wide range of nutrients and food groups commonly consumed in East Africa, which not only increased the representativeness and accuracy of the results but also demonstrated methodological forethought and attention to detail on the part of the researchers. However, this approach has limitations on standardized data with different methods of nutrient measurement and units consistent in food ingredients. A comparison of the total nutrient intake with the harmonized average requirement (H-AR) for each nutrient, except for energy, carbohydrate, and dietary fibre where the national values were used as reference based on the age group. We defined the inadequacy of intake if the total nutrient intake was below the reference values. **Table 1** presents the H-AR and the national reference values.

## Data analysis

We determined the amount of fortification vehicles, such as oil, wheat flour, sugar from home-made recipes, and sugar from industrially made foods or beverages, consumed in grams. We used the fortificant values in the TFCT and Uganda Food Composition Table (UFCT) to estimate the prevalence of inadequate vitamin A in the presence of fortified oil and the prevalence of vitamins and minerals in the presence of fortified wheat flour (vitamin B12, folate, iron, and zinc).

In **Table 2**, the additional intakes of vitamins or minerals from fortified food were calculated by multiplying the amount of fortified food consumed by the amount of nutrients or minerals added during fortification. The East African Standards for fortification and accounted for the percentage of losses between production and consumption. A 30% loss at the household level accounted for vitamin A in fortified oil and a 15% loss for vitamin B12 and folate in fortified wheat flour. The nutrient content of each food item were separately determined using TFCT and UFCT. The amount consumed, its nutrient content, and the USDA nutrient retention factor, specific to the cooking method employed, were subjected to multiplication and subsequent division by 100. This process resulted in the determination of nutrient

**Table 1. Recommended Daily Allowance (RDA) and harmonized average requirement values by age group.**

| Nutrients | Adolescents Aged 15–19 Years | Women Aged 20–49 Years |
|---|---|---|
| **Macronutrients (RDA)** * | | |
| Energy (kilocalorie [kcal]) | 2200 | 2100 |
| Protein (g) (H-AR) | (0.8 X 60 kilogram (kg)** = 48 g/day) | (0.8 X 70 kg*** = 56 g/day) |
| **Micronutrients (H-AR Values)** | | |
| Vitamin A retinol activity equivalent (RAE) (microgram [µg]) | 490 | 490 |
| Vitamin B1—Thiamine (milligram [mg]) | 0.9 | 0.9 |
| Vitamin B2—Riboflavin (mg) | 1.4 | 1.3 |
| Vitamin B3—Niacin (mg) | 11 | 11 |
| Vitamin B6 (mg) | 1.3 | 1.3 |
| Vitamin B12 (µg) | 2 | 2 |
| Vitamin C (mg) | 75 | 80 |
| Vitamin E (mg) | 12 | 12 |
| Copper (µg) | 685 | 700 |
| Folate dietary folate equivalent (µg) | 250 | 250 |
| Iron (mg) (low absorption—5 percent) | 22.4 | 22.4 |
| Zinc (mg) (unrefined diet, 1200 mg phytate/day in the diet) | 10.2 | 10.2 |

* Ryan-Harshman and Aldoori 2006

** The average weight of women in Tanzania aged 15 to 19 years old (MoH 2023)

*** the average weight of women in Tanzania aged 20–49 years old (MoH 2023)

intake measured in grams. The nutrient gap at Quartile 1 (Q1) was obtained by subtracting the usual nutrient intake at Q1 from the Abs(H-AR value). Then, the nutrient gap at Q1 was expressed as a percentage of the H-AR using the following formula: [(absolute magnitude nutrient gap at Q1/H-AR value) * 100]. The RDA value was used instead of the H-AR value for energy, as the H-AR value was not available. The probability of sufficient intake of essential

**Table 2. Amounts of fortified vitamins and minerals added at the factory and estimated percentage losses at the household level.**

| Nutrients | Added at the Factory (mg/100 g) | Estimated Losses at the Household Level |
|---|---|---|
| **Vegetable Oil** | | |
| Vitamin A | 2.5 | 30% |
| **Wheat Flour** | | |
| Vitamin B1—Thiamine | 0.86 | 0% |
| Vitamin B2—Riboflavin | 0.4945 | 0% |
| Vitamin B3—Niacin | 4.945 | 0% |
| Vitamin B6—Pyridoxine | 0.559 | 0% |
| Vitamin B12—Cyanocobalamin | 0.00129 | 15% |
| Folate—Folic acid | 0.258 | 15% |
| Iron—Ferrous fumarate | 3.44 | 0% |
| Zinc—Zinc oxide | 4 | 0% |

**Source:** As specified in the 2008 Uganda Food Consumption Survey (Harvey, Rambelson, and Dary 2010)

nutrients (such as energy, protein, and 12 micronutrients) was analyzed using SAS Studio version 9.4 (SAS Institute Inc. 2013). To evaluate iron intake, the Simulating Intake of Micronutrients for Policy Learning and Engagement (SIMPLE) SAS macro developed by the University of California, Davis was utilized due to the asymmetrical distribution of iron requirements.

## Results

Five hundred NPNL WRA, with a mean age of 32.38 years and a standard deviation of 9.95, were recruited from the Mbeya region in Tanzania for the study. A comprehensive overview of the socio-economic and demographic characteristics of the study population is presented in **Table 3**. It is noteworthy that 12.6% of the participants were aged 15–19 years, while 15.4% were in the age range of 35–39 years. Additionally, approximately 70% had attained primary education, 60.8% were self-employed, 58.3% had given birth to between one and four children, and 18.2% fell within the lowest household wealth index category. Notably, 24.8% of the participants exhibited overweight status as defined by their body mass index (BMI).

In **Table 4**, it is evident that cereal, cereal products, and cereal-based dishes were commonly served 2–3 times (77.6%) among NPNL WRA. Moreover, around 80% of NPNL WRA did not consume fruits, and 50.4% did not consume pulses, nuts, and seeds. **Table 5** shows a comprehensive list of 137 different food items. Out of these, 71 are directly reported as single foods, and the remaining 66 are distinct ingredients extracted from recipes. Among the recipes, 87 involve the use of oil (vegetable oil, sunflower oil, or palm oil), 15 are homemade recipes that use sugar, 9 are industrial recipes that use sugar, and 13 recipes use wheat flour.

**Table 6** shows the consumption of three different types of food that used for large-scale fortification. All women reported consuming oil, with consumption levels in the first and third quartiles being less than 36gm and 37gm, respectively. Consumption levels were similar in both urban and rural areas, with the mean (median) consumption being 36.54gm (36.50gm) and, 36.50gm (36.45gm) respectively. Our findings indicate that slightly over 20% of women (110 out of 500) consumed food containing wheat flour, with the Q1 value being less than 70.2gm and the Q3 value being less than 209.8gm. More urban women (31%) reported consuming wheat flour than rural women (17%), with urban consumers also reporting significantly higher consumption levels than their rural counterparts in both the Q1 (73.4gm versus 57.9gm) and Q3 (284.5gm versus 142.2gm), Levene test; *P-value*>0.05.

**Table 7** displays the median, Q1, and Q3 for the typical daily nutrient intakes of all women categorized by their place of residence, i.e., urban and rural areas. The survey found that the median energy intake among all the women was 2,169.9 kcal, and there was no difference in the intake between women residing in urban and rural areas (2,173 kcal versus 2,169.2 kcal). Women in urban areas had higher values in Q1 and Q3 than those in rural areas. The median protein intake was 63.5gm, almost the same for urban and rural women. The Q3 intake of usual energy was higher than the expected energy expenditure for women in both urban and rural areas. The Q3 intake was 6.5mg for vitamin E, 17.02mg for iron, and 1.7g for copper. These values were similar for urban and rural women and were below the recommended H-AR thresholds.

**Table 8** presents the percentage of participants in this study who had insufficient intake for each nutrient. The prevalence of inadequacy ranged from 2.6% for protein to 90.6% for zinc. The highest prevalence of inadequate intake was found for zinc (90.6%), vitamin E (89.0%), iron (80.0%), vitamin B12 (78.8%), and riboflavin (53.0%). About one-third of the women lacked sufficient amounts of energy, folate, and thiamine. Additionally, one-fourth of them were deficient in vitamin A, niacin, vitamin C, and vitamin B6. We calculated both the

**Table 3. The socio-economic and demographic characteristics of the study population (n = 500).**

| Variable | n (%) |
| --- | --- |
| **Age Group** | |
| 15–19 years | 63 (12.6) |
| 20–24 years | 77 (15.4) |
| 25–29 years | 66 (13.2) |
| 30–34 years | 69 (13.8) |
| 35–39 years | 77 (15.4) |
| 40–44 years | 74 (14.8) |
| 45–49 years | 74 (14.8) |
| **Education Level** | |
| No school | 7 (1.4) |
| Primary school | 315 (63) |
| Secondary school | 108 (21.6) |
| Higher education | 21 (4.2) |
| Did not attend | 49 (9.8) |
| **Marital Status** | |
| Married | 196 (39.2) |
| Unmarried | 304 (60.8) |
| **Occupation** | |
| Employed | 6 (1.2) |
| Self employed | 304 (60.8) |
| Not employed | 167 (33.4) |
| Other | 23 (4.6) |
| **Number of Live Births** | |
| 1–4 | 290 (58.0) |
| 5–9 | 112 (22.4) |
| None | 98 (19.6) |
| **Wealth Index** | |
| Lowest quintile | 91 (18.2) |
| Second quintile | 93 (18.6) |
| Third quintile | 93 (18.6) |
| Fourth quintile | 93 (18.6) |
| Highest quintile | 93 (18.6) |
| **Residence** | |
| Rural | 318 (63.6) |
| Urban | 182 (36.4) |
| **Council** | |
| Chunya | 17 (3.4) |
| Kyela | 68 (13.6) |
| Mbarali | 101 (20.2) |
| Mbeya City | 135 (27.0) |
| Mbeya | 66 (13.2) |
| Rungwe | 113 (22.6) |
| **BMI [kilogram /metres$^2$]** | |
| Underweight ($<$18.5) | 24 (4.8) |
| Normal (18.5 to $<$ 25) | 254 (50.8) |
| Overweight (25.0 to $<$30) | 124 (24.8) |
| Obese (30.0 or higher) | 98 (19.6) |

**Table 4. Distribution of food groups consumed per number of servings among non-pregnant non-lactating women aged 15–49 years in Mbeya.**

| | Number of Servings | | | | | | | |
| --- | --- | --- | --- | --- | --- | --- | --- | --- |
| | Not consumed | | Once | | 2–3 | | = >4 | |
| Food group | n | % | n | % | n | % | n | % |
| Cereal, cereal products and cereal-based dishes | 0 | 0.00 | 15 | 2.41 | 484 | 77.69 | 124 | 19.90 |
| Meat, poultry, eggs and fish | 296 | 47.51 | 197 | 31.62 | 125 | 20.06 | 5 | 0.80 |
| Pulses, nuts and seeds | 338 | 54.25 | 201 | 32.26 | 84 | 13.48 | 0 | 0.00 |
| Fruits | 499 | 80.10 | 92 | 14.77 | 29 | 4.65 | 3 | 0.48 |
| Vegetables and vegetable dishes | 240 | 38.52 | 233 | 37.40 | 149 | 23.92 | 1 | 0.16 |
| Sugar and Oil | 117 | 18.78 | 142 | 22.79 | 269 | 43.18 | 91 | 14.61 |
| Unhealthy foods | 444 | 71.27 | 148 | 23.76 | 30 | 4.82 | 1 | 0.16 |

absolute and the percentage of H-AR magnitude of nutrient gaps in Q1 to illustrate the severity of nutrient inadequacy among the women in our study. Q1's absolute nutrient gap ranged from 0.2 mg for thiamine to 70.1 kcal for energy. The Q1 magnitude of the nutrient gap was highest for vitamin E (50.7% of H-AR, 6.1 mg), vitamin C (42.3% of H-AR, 32.8 mg), and riboflavin (40.9% of H-AR, 0.6 mg). Between 30% and 40% of H-AR were for niacin, vitamin B12, and iron. For vitamin A, the absolute nutrient gap was 159.9μg, with a magnitude gap of 32.6% of H-AR. For vitamin B6, the absolute nutrient gap was 0.3mg, with a magnitude gap of 26.5% of H-AR. For folate, the absolute nutrient gap was 40.5μg, with a magnitude gap of 16.2% of H-AR. Finally, for zinc, the absolute gap was 1mg, with a magnitude gap of 9.8% of H-AR.

Table 9 compares the median usual nutrient intake using the TFCT and UFCT. Levene's test was used for each nutrient's median usual consumption. When only non-fortified foods (NFF) or diet is considered, the median usual intake of energy intake, vitamin A, riboflavin, niacin, vitamin B6, vitamin C, and folate showed significant differences between TFCT and UFCT (at a *P-value* of <0.05). However, when fortified foods are considered, the median usual intake of vitamin A (RAE), vitamin C, and iron showed significant differences between TFCT and UFCT (at a *P-value* of <0.05). On other nutrients, there were no significant differences in the median usual intake between TFCT and UFCT.

In Table 10, the percentage of people who don't get enough vitamin A under three different scenarios: 1) only through their diet, without fortified foods, using the TFCT, 2) through their

**Table 5. Numbers of recipes/Single foods by list type.**

| Recipes/Food Items | Number |
| --- | --- |
| **Fortification Vehicles and Sugar** | |
| Recipes containing **oil** | 87 |
| Recipes containing **wheat flour** | 13 |
| Homemade recipes containing **sugar** | 15 |
| Industrially made foods [a] containing **sugar** | 9 |
| **General Recipes/Single Foods** | |
| Total recipes | 113* |
| Total reported single foods | 71 |
| Total single foods extracted from recipes | 66 |
| **Total single foods** | **137** |

[a] biscuit, bread—brown, bread—roll, bread—white, candy, chocolate, ice cream, and soda

* The number of recipes increased when we reconstructed the recipes based on the actual ingredients reported by women

**Table 6. The consumption of food considered by fortification vehicle and sugar (Homemade and Industrially Made) among the NPNL WRA of each food.**

| Food Item | Residence | | | | | | | | Total | | | |
|---|---|---|---|---|---|---|---|---|---|---|---|---|
| | Rural (N = 319) | | | | Urban (N = 181) | | | | (N = 500) | | | |
| | n (%) | Mean (g) Median (g) | Q1 (g) | Q3 (g) | n (%) | Mean (g) Median (g) | Q1 (g) | Q3 (g) | n (%) | Mean (g) Median (g) | Q1 (g) | Q3 (g) |
| Oil[1] | 319 (100) | 36.50 | 35.99 | 37.02 | 181(100) | 36.54 | 36.12 | 37.08 | 500 (100) | 36.51 | 36.01 | 37.04 |
| | | 36.45 | | | | 36.50 | | | | 36.47 | | |
| Wheat flour[2] | 53 (17) | 131.29 | 57.89 | 142.23 | 57(31) | 234.07 | 73.68 | 284.45 | 110 (25) | 184.55 | 70.23 | 209.8 |
| | | 86.84 | | | | 142.23*** | | | | 110.53 | | |
| Sugar—Home[3] | 319 (100) | 63.59 | 36.91 | 74.39 | 181 (100) | 60.55 | 38.06 | 69.06 | 500 (100) | 62.49 | 37.39 | 72.7 |
| | | 51.54 | | | | 52.42 | | | | 51.66 | | |
| Sugar—Industrial[4] | 28 (9) | 27.27 | 18.30 | 37.55 | 29 (16) | 21.80* | 18.30 | 24.76 | 57 (11) | 24.49 | 18.30 | 33.0 |
| | | 19.89 | | | | 18.30** | | | | 19.10 | | |

* $p$-value <0.05 for Levene test centred at the mean (i.e., W0), ** $p$-value <0.05 for Levene test at the median (i.e., W50), *** $p$-value <0.01 for Levene test at the median (i.e., W50)

[1] **Oil** includes vegetable oil and palm oil, extracted from a long list of foods prepared with oil. The analysis considered usual intake based on the overall trend.

[2] **Wheat flour** includes wheat flour and food made from wheat flour. The analysis considered the observed grams consumed by each reported respondent.

[3] **Sugar—home** includes sugar extracted from a long list of foods, which are non-industrially prepared with sugar. The analysis considered usual intake based on the overall trend.

[4] **Sugar—industrial** includes sugar extracted from biscuit, bread—brown, bread—rolls, bread—white, candy, chocolate, ice cream, and soda. The analysis considered the observed grams consumed by each reporting respondent.

diet and fortified oil, using the TFCT method, and 3), using the UFCT. When looking at diet alone, the median vitamin A intake was 772.3μg. As shown in Tables 7 and 8, around 37% of people had an inadequate intake of vitamin A. However, when oil was introduced into their diet using the TFCT, the median vitamin A intake increased to 1848.9μg. When the UFCT was used, the median vitamin A intake increased to 2541.1μg. As a result, the prevalence of inadequate vitamin A intake decreased to 7.4% and 5.6%, respectively, in the last two scenarios. This finding highlights the benefits of consuming oil fortified with vitamin A.

Table 11 shows the significant reduction in inadequate intake of vitamin B12 and folate due to the fortification of wheat flour with the Tanzania Food and Drugs Authority (TFDA) formula using a Levene test. Before fortification, the median usual intake of folate was 398.2μg. However, after fortification, it increased to 493.5μg, which is a statistically significant increase at $P$-value <0.01, and the prevalence of inadequate intake decreased from 32.6 to 25.8 percent. Similarly, the median usual intake of vitamin B12 increased from 1.4μg to 2.6μg, and the prevalence of inadequate intake decreased from 78.8 to 0.4 percent, which is also statistically significant at $P$-value <0.01. For zinc, the median usual intake increased from 9.63mg to 10.5mg, which resulted in a substantial decrease in the prevalence of inadequate zinc intake from 90.6 percent to 27.8 percent. It is important to note that the wheat flour fortification formula in Tanzania does not include thiamine, riboflavin, niacin, and vitamin B6, and the inadequacy of all these micronutrients, although mild, was around 30–40 percent.

## Discussion

This study highlights a higher prevalence of inadequate micronutrient intake, such as zinc, vitamin E, iron, vitamin B12, and riboflavin, among WRA NPNL in the Mbeya Region of Tanzania. Our findings show that the insufficient intake of these micronutrients can be attributed to lower consumption of meat, dairy products, fish, poultry, whole grains, and dark green leafy vegetables [24–27]. Shreds of evidence have shown that consuming vitamin C-rich foods like

**Table 7. Usual macronutrient and micronutrient intake by place of residence.**

| Nutrient (Measurement)* | H-AR/RDA | | Residence | | | | | | Total | | |
|---|---|---|---|---|---|---|---|---|---|---|---|
| | Adolescent 15–19 Years Old | Women 20–49 Years Old | Rural (N = 319) | | | Urban (N = 181) | | | (N = 500) | | |
| | | | Mean [Median] | Q1 | Q3 | Mean [Median] | Q1 | Q3 | Mean [Median] | Q1 | Q3 |
| Energy (kcal) | 2200 | 2100 | 2171.8 [2169.2] | 2069.9 | 2270.2 | 2176.7 [2173.5] | 2091.2 | 2255.1 | 2173.6 [2169.9] | 2079.9 | 2263.5 |
| Protein (g) | 48 | 56 | 63.6 [63.5] | 59.8 | 67.2 | 63.8 [63.7] | 60.6 | 66.7 | 63.7 [63.5] | 60.2 | 67.0 |
| Vitamin A (μg [RAE]) | 490 | 490 | 1128.6 [733.4] | 324.3 | 1380.3 | 1147.1 [804.2] | 343.1 | 1505.5 | 1135.3 [772.3] | 330.1 | 1430.3 |
| Thiamine (mg) | 0.9 | 0.9 | 1.1 [1.3] | 0.7 | 1.4 | 1.1 [1.3] | 0.7 | 1.4 | 1.1 [1.3] | 0.7 | 1.4 |
| Riboflavin (mg) | 1.4 | 1.3 | 1.7 [1.2] | 0.8 | 2.7 | 1.7 [1.2] | 0.8 | 2.7 | 1.7 [1.2] | 0.8 | 2.7 |
| Niacin (mg) | 11 | 11 | 10.4 [12.4] | 7.3 | 13.6 | 10.4 [12.5] | 6.7 | 13.6 | 10.4 [12.4] | 7.3 | 13.6 |
| Vitamin B6 (mg) | 1.3 | 1.3 | 1.2 [1.5] | 1.0 | 1.6 | 1.2 [1.5] | 1.0 | 1.6 | 1.2 [1.5] | 1.0 | 1.6 |
| Vitamin B12 (μg) | 2 | 2 | 2.1 [1.4] | 1.3 | 1.5 | 2.5 [1.4] | 1.3 | 2.2 | 2.2 [1.4] | 1.3 | 1.5 |
| Vitamin C (mg) | 75 | 80 | 78.7 [102.8] | 46.6 | 111.1 | 78.7 [105.2] | 42.5 | 110.9 | 78.7 [103.9] | 44.7 | 111.0 |
| Vitamin E (mg) | 12 | 12 | 7.7 [6.2] | 5.9 | 6.5 | 7.8 [6.2] | 5.9 | 6.5 | 7.7 [6.2] | 5.9 | 6.5 |
| Copper (g) | 0.685 | 0.7 | 1.7 [1.7] | 1.6 | 1.7 | 1.7 [1.6] | 1.6 | 1.7 | 1.7 [1.7] | 1.6 | 1.7 |
| Folate (μg) | 250 | 250 | 328.5 [397.8] | 209.7 | 419.6 | 328.7 [401.6] | 208.2 | 422.1 | 328.6 [398.2] | 209.6 | 421.0 |
| Iron (mg) | 22.4 | 22.4 | 20.5 [16.0] | 15.4 | 17.2 | 21.0 [16.1] | 15.5 | 16.9 | 20.7 [16.1] | 15.4 | 17.0 |
| Zinc (mg) | 10.2 | 10.2 | 9.2 [9.6] | 9.2 | 9.9 | 9.2 [9.7] | 9.2 | 10.0 | 9.2 [9.6] | 9.2 | 10.0 |

* We used the TFCT to estimate the nutrient intake and did not consider the effect of fortified food

oranges, papaya, cabbage, and green leafy vegetables can help the body absorb non-heme iron found in foods such as nuts, beans, or amaranth leaves. These foods are frequently consumed by WRA NPNL in Mbeya [28].

The study took place during the dry season, and the seasonal variations could significantly impact the outcomes of the results of the 24-hour recall survey. Certain fruits or vegetables could be plentiful during specific times but scarce or unavailable at other times. This variability could create bias in regular dietary intake, resulting in inaccurate nutrient intake [22, 23]. However, the NCI usual intake model takes this issue into account and adjusts for days when dietary data might be missing, recognizing that people don't eat the same way every day and that food availability changes with the seasons [22, 23].

The outcomes of our research align with various studies conducted globally. Goh et al. (2023) reported that NPNL WRA exhibited inadequate dietary intake of essential micronutrients such as iron, zinc, and vitamin B12 [26]. Similarly, Otunchieva et al. (2022) documented that WRA in low-income countries experienced micronutrient deficiencies [16]. Furthermore, Sharma et al. (2020) unveiled that a significant proportion of WRA worldwide consumed less than the recommended dietary intake of iron [18]. In southeastern Nigeria, Onyeji and Sanusi (2022) identified low micronutrient intake among WRA, with urban women having slightly

**Table 8. Usual nutrient intake, prevalence of intake below the H-AR, and magnitude of nutrient gap at Q1 (N = 500, Subsample n = 120 for Percent below H-AR Analysis).**

| Nutrient* (Measurement) | Median | Q1 | Q3 | Prevalence of Inadequate Intake (percent below H-AR) | Absolute Magnitude of Nutrient Gap at Q1[±] | Magnitude of Nutrient Gap at Q1 (% H-AR)[ǂ] |
|---|---|---|---|---|---|---|
| Energy (kcal) | 2169.88 | 2079.90 | 2263.54 | 34.4 | 70.1 | 3.3 |
| Protein (g) | 63.52 | 60.21 | 66.99 | 2.6 | - | - |
| Vitamin A (µg [RAE]) | 772.26 | 330.12 | 1430.32 | 37.0 | 159.9 | 32.6 |
| Thiamine (mg) | 1.23 | 0.70 | 1.41 | 34.8 | 0.2 | 22.8 |
| Riboflavin (mg) | 1.23 | 0.79 | 2.70 | 53.0 | 0.6 | 41.9 |
| Niacin (mg) | 12.48 | 7.26 | 13.59 | 44.4 | 3.7 | 34.0 |
| Vitamin B6 (mg) | 1.47 | 0.96 | 1.59 | 45.2 | 0.3 | 26.5 |
| Vitamin B12 (µg) | 1.37 | 1.29 | 1.53 | 78.8 | 0.7 | 35.5 |
| Vitamin C (mg) | 103.93 | 44.72 | 111.00 | 45.6 | 32.8 | 42.3 |
| Vitamin E (mg) | 6.18 | 5.91 | 6.51 | 89.0 | 6.1 | 50.7 |
| Copper (g) | 1.65 | 1.58 | 1.73 | 0.0 | - | - |
| Folate (µg) | 398.18 | 209.55 | 420.99 | 32.6 | 40.5 | 16.2 |
| Iron (mg) | 16.05 | 15.39 | 17.02 | 80.0 | 7.0 | 31.3 |
| Zinc (mg) | 9.63 | 9.21 | 9.96 | 90.6 | 1.0 | 9.8 |

*We used the TFCT to estimate the nutrient intake and did **not** consider the effect of fortified food

[±] Calculated as H-AR value—Absolute nutrient intake at Q1 (in absolute value). We used the RDA value for energy.

[ǂ] Calculated as absolute nutrient gap/H-AR or RDA (in percentage)

higher energy and protein intake than their rural counterparts [15]. These findings are consistent with those of another study conducted among WRA in Vietnam by Nguyen et al. (2013) [14].

**Table 9. Comparison of nutrient intake from reported food non-fortified and those considered fortified according to the Tanzania and Uganda food composition tables.**

| Nutrient (Measurement) | Non-Fortified Foods (N = 500) | | | | Fortified Foods (N = 500) | | | |
|---|---|---|---|---|---|---|---|---|
| | TFCT | | UFCT | | TFCT | | UFCT | |
| | Median (g) | + IQR | Median (g) | + IQR | Median (g) | + IQR | Median (g) | + IQR |
| Energy (kcal) | 2169.88 | 183.98 | **2238.55\*** | 193.22 | 2260.96 | 243.11 | 2265.76 | 232.12 |
| Protein (g) | 63.52 | 6.79 | 63.90 | 7.09 | 66.26 | 7.76 | 66.28 | 7.83 |
| Vitamin A (µg [RAE]) | 772.26 | 1102.1 | **1499.78[a]** | 138.55 | 1848.91 | 155.70 | **2541.06\*** | 1871.1 |
| Thiamine (mg) | 1.23 | 0.72 | 1.14 | 0.64 | 1.25 | 0.66 | 1.23 | 0.75 |
| Riboflavin (mg) | 1.23 | 1.91 | **1.08[a]** | 0.07 | 1.23 | 1.91 | 1.16 | 0.07 |
| Niacin (mg) | 12.48 | 6.35 | **13.09[a]** | 6.73 | 12.68 | 6.96 | 12.80 | 6.90 |
| Vitamin B6 (mg) | 1.47 | 0.64 | **1.90[a]** | 0.95 | 1.49 | 0.98 | 1.54 | 0.91 |
| Vitamin B12 (µg) | 1.37 | 0.24 | 1.73 | 0.32 | 2.63 | 0.34 | 2.63 | 0.34 |
| Vitamin C (mg) | 103.93 | 66.37 | **78.32[a]** | 56.94 | 90.40 | 60.20 | **78.32\*** | 61.98 |
| Vitamin E (mg)[†] | 6.18 | 0.60 | —— | —— | 6.18 | 0.60 | —— | —— |
| Copper (g)[†] | 1.65 | 0.15 | —— | —— | 1.65 | 0.15 | —— | —— |
| Folate (µg) | 398.18 | 211.60 | **411.79[a]** | 214.96 | 493.46 | 288.53 | 493.81 | 287.40 |
| Iron (mg) | 16.05 | 1.64 | 16.58 | 1.96 | 17.55 | 2.70 | **19.82\*** | 4.30 |
| Zinc (mg) | 9.63 | 0.76 | 9.68 | 1.12 | 9.64 | 0.777 | 9.69 | 1.13 |

[†] Vitamin E and copper were not available in the Uganda FCT

[a] $p$-value $<0.05$ non-fortified TFCT vs. UFCT; and [b] $p$-value $<0.05$ fortified TFCT vs. UFCT. The study uses $p$-values computed from the Levene's test using the median.

**Table 10. Usual intake and predicted prevalence of inadequate vitamin A intake in the presence of fortification, Levene test.**

| Predicted Scenario | Vitamin A Intake (N = 500) | |
|---|---|---|
| | Median (µg) | Inadequacy (%) |
| Diet alone—NFF TFCT | 772.26[a,b] | 37%[a,b] |
| Diet plus fortified oil—FF TFCT | 1848.91 | 7.4% |
| Diet plus fortified oil—FF UFCT | 2541.07 | 5.6% |

[a] p-value< 0.001 non-fortified TFCT vs. fortified TFCT
[b] p-value< 0.001 non-fortified TFCT vs. fortified UFCT

Literature emphasizes the importance of examining the nutrient gap in Q1, as it is the gap that needs to be filled with micronutrient-delivering interventions [29]. This study explores the significance of multi-faceted interventions for addressing inadequate intake of nutrients, especially vitamin E, iron, vitamin B12, riboflavin, and vitamin C. Our study shows a relatively low level of zinc inadequacy (9.8 percent of H-AR), pointing to the role of food fortification and improved dietary practices to significantly reduce the high prevalence of inadequate zinc intake (91.2 percent) [29].

Our survey found that people in the NPNL WRA group consumed more vitamin A-rich foods, such as carrots, sweet potatoes with orange flesh, and green leafy vegetables like pumpkin leaves, spinach, and amaranth leaves. However, despite this, vitamin A deficiency remains a concern. Therefore, fortifying foods with vitamin A could be a potential solution to prevent deficiency [29–31].

All the participants surveyed reported consuming oil, with similar intake in both urban and rural areas. This is because nearly every recipe they used included oil as a basic ingredient in food preparation. Vegetable oil or palm oil was the most commonly used oils, extracted from a variety of foods. The study found that women in rural areas consumed higher amounts of sugar than those in urban areas [32–36]. The sugar intake included all sugars added to foods or drinks, whether industrially manufactured or added in homemade foods, as well as sugars added in fruit juices and fruit juice concentrates.

The study also assessed the consumption of fortified foods, including edible oil and wheat flour. In 2015, the consumption of edible oil among WRA was 22 g/day, and for wheat flour, it was 162 g/day [7]. However, these numbers were computed using a different methodology than the one used in this study. The study also found a low coverage, use, and intake of wheat

**Table 11. Usual intake and predicted prevalence of inadequate micronutrient intake in the presence of fortification, Levene test.**

| Nutrient Added | NFF TFCT (N = 500) | | FF TFCT (N = 500) | |
|---|---|---|---|---|
| | Median | Inadequacy (%) | Median | Inadequacy (%) |
| Thiamine (mg) | 1.23 | 34.8% | 1.25 | 34.8% |
| Riboflavin (mg) | 1.23 | 53.0% | 1.23 | 53.0% |
| Niacin (mg) | 12.48 | 44.4% | 12.68 | 44.4% |
| Vitamin B6 (mg) | 1.47 | 45.2% | 1.49 | 45.2% |
| Vitamin B12 (µg) | 1.37 | 78.8% | 2.63[a] | 0.4% [a] |
| Folate (µg) | 398.18 | 32.6% | 493.46[a] | 25.8% |
| Iron (mg) | 16.05 | 80.0% | 17.55 | 76.2% |
| Zinc (mg) | 9.63 | 90.6% | 10.57[a] | 27.8%[a] |

[a] p-value <0.001, NFF TFCT vs. FF TFCT

flour, although the average quantity of intake in the consumers was higher among urban women.

The study further highlighted that the consumption of fortified foods (oil and wheat flour) led to increased intake of vitamins A, B12 and zinc, and decreased the prevalence of inadequate intake compared to the values of non-fortified foods. As part of the data-cleaning process, the study developed various tools such as a comprehensive list of recipes with their ingredient breakdowns and the harmonized conversion of local measurements to standard units. These tools can potentially improve and refine standardized Tanzania 24-hour recall food intake survey instruments, including multi-pass questionnaires, a food photo book, and a Kobo tablet-based data entry template.

Our study has certain limitations in the collection and analysis of data. Nevertheless, we have taken measures to reduce their impact on our results to the maximum possible extent. Firstly, we did not collect information on the bioavailability of different micronutrients affected by phytate, nor did we gather data on the specific brands of oil or wheat flour (vehicles for fortification) consumed by the women. In our analysis, we employed the government-mandated fortificant quantities, as opposed to the actual fortification levels, to conduct a simulation. The objective was to estimate the potential impact of fortified oil on insufficient vitamin A intake and fortified wheat flour on insufficient vitamin B12, folate, iron, and zinc intake. Furthermore, we addressed thiamine, riboflavin, niacin, and vitamin B6 to highlight their persistently high inadequacies attributable to the absence of these micronutrients in fortified wheat flour. The simulation analysis adhered to the recommended range of values for micronutrient fortificants, including vitamin A and others.

Second, we discovered some inaccuracies in the nutrient content values of TFCT and implemented several measures to address them. To correct the caloric content, we used the 4:4:9:2 rule, which involves multiplying the protein, carbohydrate, fat, and total fiber content in grams by 4, 4, 9, and 2 kcal/gram, respectively. We also examined the bioconversion factor of pro-vitamin A carotenoid to retinol in TFCT and UFCT. Finally, we conducted a comparative analysis between TFCT and UFCT to identify potential discrepancies or confirm the validity of intake inadequacy findings. However, we couldn't compare the vitamin E and copper content of TFCT and UFCT because there was no information on these two elements in UFCT.

Third, our findings were based on the consumption data of NPNL WRA in the Mbeya Region only. Therefore, it would be inappropriate to generalize these findings to comment on the dietary patterns, behaviours, and nutrient intakes of the general population or other population sub-groups. We also noticed a specific limitation of zero value or missing data management in the episodic SAS macro for riboflavin and vitamin B12. The episodic model is designed to estimate the usual intake for nutrients or foods that are infrequently consumed, and it assumes that missing data or zero value is either completely random or systematic. To address this limitation, we conducted various scenarios. In our preliminary findings, we followed the NCI guideline's advice and adjusted zero values to half of the minimum values. We also set the number of bootstrap tests at 100. However, our preliminary results showed that 100 percent of riboflavin values exceeded the H-AR threshold of 1.4, while 100 percent of vitamin B12 values fell below the threshold of 2. Consequently, we opted to retain the zero values, but we doubled the number of bootstrap tests from 100 to 200 and amplified the weight coefficient by a factor of 100. These adjustments resulted in a realistic prevalence of inadequacy intake for the nutrients in this report. However, it is important to approach the interpretation and use of these results with caution, as our data may not strictly adhere to the completely random pattern required for the episodic SAS macro. This deviation could potentially introduce bias and lead to less robust estimates.

## Conclusion

Our study has shown a highly prevalent of inadequate intake of micronutrients among NPNL WRA in Mbeya region, Tanzania. The insufficient intake of these micronutrients can be attributed to lower consumption of meat, dairy products, fish, poultry, whole grains, and dark green leafy vegetables. The findings highlight the need for increased attention towards the intake of vitamin E, iron, and vitamin B12, followed by riboflavin and vitamin C, based on the prevalence of inadequate intake. However, for all cases, the highest nutrient gaps is lower than 50 percent H-AR for these micronutrients and lower than 35 percent for the other vitamins of the B-complex (thiamine, niacin, vitamin B6, and folate). The low inadequacy of zinc intake could be easily rectified due to its very low magnitude nutrient gap. Moreover, our study suggests that the prevalence of inadequate intake of vitamin A, vitamin D and vitamin E can be significantly reduced among women given the usual intake of oil. In this context, oil emerges as a promising candidate for fortification. However, fortified wheat flour would have a limited impact on the nutrient gaps observed due to its limited reach among NPNL WRA in Mbeya. Nonetheless, further validation analysis of episodic nutrient consumption is still needed to enhance the reliability and robustness of the usual intake values, particularly for riboflavin and vitamin B12, within the context of Mbeya region, Tanzania.

## Supporting information

**S1 Checklist. STROBE checklist.**
(DOCX)

## Author Contributions

**Conceptualization:** Nyamizi J. Ngassa, Adam Hancy, Rose Mpembeni, Theresia Jumbe, Germana H. Leyna.

**Data curation:** Adam Hancy, Esther J. Kabula, Erick Killel, Rose Mpembeni.

**Formal analysis:** Nyamizi J. Ngassa, Ray M. Masumo, Adam Hancy, Esther J. Kabula, Jackline Nusurupia, Adeline Munuo, Hoyce Mshida, Rose Mpembeni, Elizabeth Lyimo, Rose V. Msaki, Theresia Jumbe, Germana H. Leyna.

**Funding acquisition:** Germana H. Leyna.

**Investigation:** Erick Killel, Jackline Nusurupia.

**Methodology:** Nyamizi J. Ngassa, Ray M. Masumo, Adam Hancy, Esther J. Kabula, Erick Killel, Jackline Nusurupia, Hoyce Mshida, Rose Mpembeni, Elizabeth Lyimo, Rose V. Msaki, Theresia Jumbe, Germana H. Leyna.

**Project administration:** Ray M. Masumo, Erick Killel, Adeline Munuo, Hoyce Mshida, Rose Mpembeni, Theresia Jumbe, Germana H. Leyna.

**Software:** Nyamizi J. Ngassa, Adam Hancy, Esther J. Kabula, Elizabeth Lyimo, Rose V. Msaki.

**Supervision:** Ray M. Masumo, Rose Mpembeni, Theresia Jumbe, Germana H. Leyna.

**Validation:** Nyamizi J. Ngassa, Adam Hancy, Esther J. Kabula, Erick Killel, Jackline Nusurupia, Hoyce Mshida.

**Visualization:** Nyamizi J. Ngassa, Esther J. Kabula, Germana H. Leyna.

**Writing – original draft:** Nyamizi J. Ngassa, Ray M. Masumo, Adam Hancy, Esther J. Kabula, Erick Killel, Jackline Nusurupia, Adeline Munuo, Hoyce Mshida, Rose Mpembeni, Elizabeth Lyimo, Rose V. Msaki, Theresia Jumbe, Germana H. Leyna.

**Writing – review & editing:** Nyamizi J. Ngassa, Ray M. Masumo, Adam Hancy, Esther J. Kabula, Erick Killel, Jackline Nusurupia, Adeline Munuo, Hoyce Mshida, Rose Mpembeni, Elizabeth Lyimo, Rose V. Msaki, Theresia Jumbe, Germana H. Leyna.

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
