## [Decision Letter · Decision Letter 0]

16 Aug 2024

PGPH-D-24-01236

Food and nutrient intake among non-pregnant, non-lactating women of reproductive age in Tanzania: A repeated 24-hour dietary recall

Dear Dr. Masumo,

Thank you for submitting your manuscript to PLOS Global Public Health. After careful consideration, we feel that it has merit but does not fully meet PLOS Global Public Health’s publication criteria as it currently stands. Therefore, we invite you to submit a revised version of the manuscript that addresses the points raised during the review process.

The manuscript has been evaluated by four reviewers, and their comments are available below and in the attached document.

The reviewers have raised a number of concerns that need attention, and request additional information and greater clarity throughout the manuscript.

Could you please revise the manuscript to carefully address the concerns raised?

We look forward to receiving your revised manuscript.

Kind regards,

Steve Zimmerman, PhD

PLOS Staff Editor

Journal Requirements:

1. You indicated that you had ethical approval for your study. In your Methods section, please ensure you have also stated whether you obtained consent from parents or guardians of the minors included in the study or whether the research ethics committee or IRB specifically waived the need for their consent.

Additional Editor Comments (if provided):

Reviewers' comments:

Reviewer's Responses to Questions

**Comments to the Author**

1. Does this manuscript meet PLOS Global Public Health’s publication criteria? Is the manuscript technically sound, and do the data support the conclusions? The manuscript must describe methodologically and ethically rigorous research with conclusions that are appropriately drawn based on the data presented.

Reviewer #1: Yes

Reviewer #2: Yes

Reviewer #3: Yes

Reviewer #4: Yes

2. Has the statistical analysis been performed appropriately and rigorously?

Reviewer #1: Yes

Reviewer #2: Yes

Reviewer #3: Yes

Reviewer #4: Yes

3. Have the authors made all data underlying the findings in their manuscript fully available (please refer to the Data Availability Statement at the start of the manuscript PDF file)?

Reviewer #1: Yes

Reviewer #2: Yes

Reviewer #3: Yes

Reviewer #4: Yes

4. Is the manuscript presented in an intelligible fashion and written in standard English?

Reviewer #1: Yes

Reviewer #2: Yes

Reviewer #3: Yes

Reviewer #4: Yes

5. Review Comments to the Author

Reviewer #1: It is a pleasure to review the paper. This article provides an in-depth analysis of the nutritional intake of non-pregnant and non-lactating women of reproductive age in Mbeya District, Tanzania. The results may be useful for the future development of nutrition policies that address the issue of inadequate dietary intake among women of reproductive age. However, I would recommend some modifications to the current manuscript.

Major issues

1. The title is inappropriate because the data come from people in only one region of Tanzania and are not nationally representative.

2. The Sample size and sampling procedures and Data Collection sections need to be written more clearly.

How was the sample size of the survey determined? Is it representative?

Only 100 people in total participated in the 24-hour dietary recall survey? 5 people in each EA (15 in total) had a second review?

Were the 24-hour dietary recall conducted on weekdays or weekends, and were the survey days the same in the 3 EAs? Because these may lead to differences in residents' food choices.

3. Data cleaning and preparation section.

Am I to understand that you investigated the proportion of each food intake and calculated the intake at a later stage? Were specific intakes investigated? If not, how do you avoid the error caused by simply using the reference chart? After all, the actual size is not the same as the printed size.

Were the survey data directly weighed/recorded by the enumerators in the households or self-reported by the respondents?

4. Data Analysis section.

I feel that Method part is too long compared to the main text, especially the Data Analysis section. The researcher spends a lot of space describing the recipes and food components tables used in the survey, as well as the treatment of nutrient content. It is recommended to simplify.

Minor comments

1. Line 43: change “15-49”to “15-49 years”.

2. Line 88: delete “(WRA)”

3. Line 93-95: “Due to the limited coverage and longer periodicity……diets of targeted populations”. This sentence feels abrupt. I do not understand the function of this sentence and how it connects to the previous and the following sentences.

4. Lines 143-144 “The survey was translated from English to Swahili and back translated for accuracy by different translators.” Do different translations lead to different understandings of the problem? Is it standardized?

5. Food matching, data quality control and validation section. The use of multiple food composition tables to supplement the sample pool was designed to be thoughtful and comprehensive. This approach ensured that the study covered a wide range of nutrients and food groups, which not only increased the representativeness and accuracy of the results, but also demonstrated methodological forethought and attention to detail on the part of the researchers. However, this design may require consideration of data consistency. How can consistency in nutrient measurement methods and units be ensured for data extracted from different sources of food composition tables? Please add.

6. Line 208: the P should be italicized P. Please also check the whole text.

7. Why did the authors identify nutrient deficiencies and compare differences in results based on two different food composition tables (TFCT and UFCT)? What was the purpose? I did not see this explored in the discussion section.

8. Some of the writing is a bit wordy. Be more concise and direct, especially by reducing the use of the first person narration. We employed, we cross-checked, We used, We extracted……

9. After revising the above points, please show the STROBE guidelines for a cross-sectional study (https://www.strobe-statement.org/checklists/).

Reviewer #2: Reviewer’s comment

It is an interesting piece of research work that highlights food and nutrient intake among NPNL WRA age in Tanzania. The study provides insights into nutrient intake that might have policy implications. However, the following issues should be considered to give a clear and comprehensive message to the readers.

1. The participants were from the seven districts of the Mbeya region of Tanzania. It may not be representative of the entire Tanzania. Thus, it is better to mention the Mbeya region in the title. You can revise the title to “Food and nutrient intake among non-pregnant, non-lactating women of reproductive age of Mbeya region in Tanzania: A repeated 24-hour dietary recall”.

2. The background, rationale, and objective of the study are missing in the abstract. It is recommended to include these points in the abstract.

3. Line #40, please add the full form with the abbreviation when it appears first and then use the abbreviation after forward. Follow this throughout the manuscript where it is applicable.

4. Lines #125-128, 7 districts, 3 EA from each district, 22 HH from each EA comes to 7*3*22 = 462. But the total number of samples is 500. So there remains confusion.

5. Lines #126-127, the authors mentioned that they used probability proportional to size (PPS). At the same time, they mentioned that an equal number of EA and HH were considered for all. More clarification is needed here.

6. Lines #130-131, please mention how one woman was selected for participation when you found more than one NPNL WRA in the same household.

7. Lines #131-132, it is not clear whether you obtained data on dietary intake for 100 participants or 500 participants.

8. Lines #132-134, how it is possible to address the effect of seasonality in dietary intake just visiting the household one week after the first visit.

9. Did you conduct any pretesting? If you did so, please give a few points on the location, how many participants, non-response rate, etc.

10. Lines #172-175, didn't you collect the information regarding the brand of edible oils they used in cooking or the brand of wheat they consumed? Then you can easily find whether they used a fortified one or a non-fortified one. Thus, you can easily avoid the underestimation or overestimation of nutrient intake.

11. It is not clear whether you collected individual dietary recall data or intake data at the household level. Please specify it.

12. Phytate affects the bioavailability of different micronutrients. Please consider this in your findings or at least discuss it as study limitations.

13. The title of the study indicates the food and nutrient intake of the study participants. But no details regarding different food group intake are available in the result section. You included nutrient intake information in detail but overlooked the food intake (only included fortified oils, wheat, and sugar). It is recommended to include information on food group intake and discuss it to justify the nutrient intake inadequacy.

14. The conclusion section includes some recommendations on food fortification rather than dietary diversification or a food-based dietary approach. It is recommended to include some findings on food group intake and make recommendations based on those findings along with fortified oils and/or wheat.

15. Lines #156-157, In case of dietary data, it is recommended to use Goldberg equation to identify outliers rather than just descriptive statistics. Using it will help to achieve more accurate outcomes.

16. It is not clear why you compare harmonized average requirement (H-AR) rather than RDA with the nutrient intake. Use of First Quartile (Q1) values rather than median also not clear. Please specify these. Also there is no reference for the harmonized average requirement (H-AR) values.

17. Table #3. Please define the Number in the Initial List & Number in the Updated List in details. As these are not clear from the footnote.

18. Table #4. Please explain the reason of using p value from the Levene test. Follow this throughout the manuscript where it is applicable.

19. Table #6. Please specify the meaning of weighted and unweighted Prevalence of Inadequate Intake (percent below H-AR), as nothing is mentioned regarding the sample weight in the manuscript.

20. Table #8. The unit of the median value of vitamin A intake should be rechecked.

21. Table #9. it is recommended to use the source (Name of the statistical test) of the p value. Follow this throughout the manuscript where it is applicable.

Reviewer #3: The manuscript presents valuable insights into the dietary intake and micronutrient status of women in Tanzania, with a thorough methodology and detailed results. However, it can be improved. See my highlights in the attached document.

Reviewer #4: The paper evaluated Food and nutrient intake among non pregnant, non lactating women of reproductive age in Tanzania. A repeated 24hr diet recall. The manuscript is informative; however, there are still several points need to be clarified or revised before publication:

Abstract:

Acronym WRA should be mentioned on line 31 (first line of abstract) rather than on line 67 (first line of introduction)

Line 51: non lactant should be non-lactating

Introduction:

Line 72: Add space between 2021(3)

Lines 82, 84, 85, 102, 119: Use WRA instead of full form

Lines 90, 103: It should be "non lactating" rather than lactating

Methods: Sampling procedure was well explained, however additional details needed about the following:

Did the authors use validated questionnaires (adopted from previous study) or were the questionnaires validated? If yes, that should be mentioned. The limitation clearly says information regarding specific brands of oils and wheat flour consumed by women is not collected. This is a big flaw and dilutes the accuracy of the date collection, data analysis and the overall results. This may affect the results of various nutrients especially for nutrient like Vitamin E which is obtained mostly from oils and B-vitamins which are present in wheat flour. How will you justify this?

Line 178-179: Sentence is not making sense. it needs to be rephrased/reworded.

Line 201: The author needs to Define Q1. It should be "Quartile 1(Q1)"

Results

Line 231- Usually SD and IQR is mentioned along with Mean and median respectively. The author has just mentioned the mean and median.

Line 233: Replace Quartile 1 and Quartile 3 with Q1 and Q3

Line 235: p value is not mentioned in the manuscript. Exact p-value should be mentioned.

Exact p values are neither mentioned in tables 4, 5, 6, 7,8,9 nor in the text. It should be mentioned either in text or in table to give the reader a clear picture of the significance.

Lines 240-241, 266, 268, 283, 286, - exact values should be mentioned if not mentioned in the table.

Table 8 only median shown. Add Interquartile range values and exact p-values.

Discussion:

Recent references can be used instead of Ref 24, 30 & 31.

Line 341- WRA can be mentioned rather than full form

Line 347: Remove "and" between Vitamin A, Vitamin B12

Line 396: lowers should be replaced with lower

Line 399: it mentions inadequate intake of Vitamin A can be significantly reduced with the usual intake of oil. Why is the author mentioning only vitamin A and not other vitamin esp Vitamin E, Vitamin D which are now in oils?

Line 402 - NPNL should be used instead of non pregnant, non lactating

6. PLOS authors have the option to publish the peer review history of their article (what does this mean?). If published, this will include your full peer review and any attached files.

**Do you want your identity to be public for this peer review?** For information about this choice, including consent withdrawal, please see our Privacy Policy.

Reviewer #1: No

Reviewer #2: No

Reviewer #3: No

Reviewer #4: No

---

## [Decision Letter · Decision Letter 1]

28 Oct 2024

PGPH-D-24-01236R1

Food and nutrient intake among non-pregnant, non-lactating women of reproductive age of Mbeya in Tanzania: A repeated 24-hour dietary recall

Dear Dr. Masumo,

Thank you for submitting your manuscript to PLOS Global Public Health. After careful consideration, we feel that it has merit but does not fully meet PLOS Global Public Health’s publication criteria as it currently stands. Therefore, we invite you to submit a revised version of the manuscript that addresses the points raised during the review process.

Reviewer 1 has requested a minor revision to clarify the decision to include two food tables in the study and ensure the readers will understand rationale for this approach.

We look forward to receiving your revised manuscript.

Kind regards,

Jennifer Tucker, PhD

Staff Editor

Journal Requirements:

Additional Editor Comments (if provided):

Reviewers' comments:

Reviewer's Responses to Questions

**Comments to the Author**

1. If the authors have adequately addressed your comments raised in a previous round of review and you feel that this manuscript is now acceptable for publication, you may indicate that here to bypass the “Comments to the Author” section, enter your conflict of interest statement in the “Confidential to Editor” section, and submit your "Accept" recommendation.

Reviewer #1: All comments have been addressed

Reviewer #4: All comments have been addressed

2. Does this manuscript meet PLOS Global Public Health’s publication criteria? Is the manuscript technically sound, and do the data support the conclusions? The manuscript must describe methodologically and ethically rigorous research with conclusions that are appropriately drawn based on the data presented.

Reviewer #1: Yes

Reviewer #4: Yes

3. Has the statistical analysis been performed appropriately and rigorously?

Reviewer #1: Yes

Reviewer #4: Yes

4. Have the authors made all data underlying the findings in their manuscript fully available (please refer to the Data Availability Statement at the start of the manuscript PDF file)?

Reviewer #1: Yes

Reviewer #4: Yes

5. Is the manuscript presented in an intelligible fashion and written in standard English?

Reviewer #1: Yes

Reviewer #4: Yes

6. Review Comments to the Author

Reviewer #1: The author provided positive feedback on my questions and suggestions, and the article now reads more clearly as a result. Only one minor comment. Before PLOS Global Public Health decides to accept this article, I would like the author to revise and explain the following issues:

In my previous opinion, I mentioned that "In Food matching, data quality control and validation section, the author used two different food composition tables (TFCT and UFCT)." I think the authors' intention was to enrich the food composition tables, but the authors did not clearly explain this in the article. I would have preferred that the authors state the reason for using two tables in the "Methods" section and explain whether it is appropriate to use data from these two different sources at the same time (Are nutrient measurement methods and units consistent in food ingredient lists from different sources?) instead of writing my original statement directly into the article (Page 7; Line 168-173).

Reviewer #4: The authors have addressed all the comments made in the first round of review

7. PLOS authors have the option to publish the peer review history of their article (what does this mean?). If published, this will include your full peer review and any attached files.

**Do you want your identity to be public for this peer review?** For information about this choice, including consent withdrawal, please see our Privacy Policy.

Reviewer #1: No

Reviewer #4: No

---

## [Decision Letter · Decision Letter 2]

14 Nov 2024

Food and nutrient intake among non-pregnant, non-lactating women of reproductive age of Mbeya in Tanzania: A repeated 24-hour dietary recall

PGPH-D-24-01236R2

Dear Dr. Masumo,

We are pleased to inform you that your manuscript 'Food and nutrient intake among non-pregnant, non-lactating women of reproductive age of Mbeya in Tanzania: A repeated 24-hour dietary recall' has been provisionally accepted for publication in PLOS Global Public Health.

Best regards,

Julia Robinson

Executive Editor

Reviewer Comments (if any, and for reference):

Reviewer's Responses to Questions

**Comments to the Author**

1. If the authors have adequately addressed your comments raised in a previous round of review and you feel that this manuscript is now acceptable for publication, you may indicate that here to bypass the “Comments to the Author” section, enter your conflict of interest statement in the “Confidential to Editor” section, and submit your "Accept" recommendation.

Reviewer #1: All comments have been addressed

2. Does this manuscript meet PLOS Global Public Health’s publication criteria? Is the manuscript technically sound, and do the data support the conclusions? The manuscript must describe methodologically and ethically rigorous research with conclusions that are appropriately drawn based on the data presented.

Reviewer #1: Yes

3. Has the statistical analysis been performed appropriately and rigorously?

Reviewer #1: Yes

4. Have the authors made all data underlying the findings in their manuscript fully available (please refer to the Data Availability Statement at the start of the manuscript PDF file)?

Reviewer #1: Yes

5. Is the manuscript presented in an intelligible fashion and written in standard English?

Reviewer #1: Yes

6. Review Comments to the Author

Reviewer #1: The authors have addressed all the comments made in the second round of review.

7. PLOS authors have the option to publish the peer review history of their article (what does this mean?). If published, this will include your full peer review and any attached files.

**Do you want your identity to be public for this peer review?** For information about this choice, including consent withdrawal, please see our Privacy Policy.

Reviewer #1: No
